# Inhibiting concentration quenching in Yb³⁺-Tm³⁺ upconversion nanoparticles by suppressing back energy transfer

Dingxin Huang ®[1,2], Feng Li[1,2], Hans Ågren[1,3] & Guanying Chen ®[1,2] ✉

Lanthanide-doped upconversion nanoparticles are promising for applications ranging from biosensing, bioimaging to solid-state lasing. However, their brightness remains limited by the concentration quenching effect of lanthanide activator ions, which greatly restricts their utility. Here, we develop a heterogeneous core–shell–shell nanostructure based on hexagonal $NaYF_4$, in which $Tm^{3+}$ activator and $Yb^{3+}$ sensitizer are separated into the core and inner shell, while the outmost shell is used to suppress surface quenching effects. We show that this design can alleviate the activator concentration quenching effect, resulting in optimal $Tm^{3+}$ concentration increasing from 1% to 8% at sub-100 W/cm² irradiance, compared with the canonical core-only $NaYF_4$:$Yb^{3+}$/$Tm^{3+}$. Moreover, under high excitation irradiance (20 MW/cm²), the optimal $Tm^{3+}$ concentration could be further increased to 50%. Mechanistic investigations reveal that the spatial separation of sensitizer and activator effectively suppresses the back energy transfer from $Tm^{3+}$ to $Yb^{3+}$, driving the increase of optimal activator concentration. These findings enhance our understanding of lanthanide concentration quenching effect, unleashing opportunities for developing bright upconverting materials.

Lanthanide-doped upconversion nanoparticles (UCNPs) have extensive ramifications for applications in areas like biosensing[1–6], bioimaging[7–14] and solid-state lasing[15–18]. UCNPs typically consist of a low-phonon-energy material as host lattice, such as hexagonal sodium yttrium fluoride (β-$NaYF_4$, ~350 cm⁻¹ phonon energy) and are doped with ytterbium ($Yb^{3+}$) ions as sensitizers. These sensitizer ions can harvest near-infrared (NIR) light and then non-radiatively transfer excitation energy to activator ions, such as thulium ($Tm^{3+}$), erbium ($Er^{3+}$) or holmium ($Ho^{3+}$), producing high-energy upconversion luminescence (UCL). Recent progress in chemical synthesis has enabled precise manipulation over the size, morphology, crystalline phase and multilayer core/shell structure, offering unprecedented opportunities for modulation of luminescence color and lifetime[19,20].

Nevertheless, it is still challenging to obtain bright upconversion due to the limited absorption/emission cross sections of lanthanide ions and most importantly, the presence of lanthanide concentration quenching effects. The concentration quenching phenomenon causes a decrease in brightness when the lanthanide dopant concentration exceeds a certain threshold. This is primarily due to the concentration-dependent nonradiative depopulation of the emitting states or the involved intermediary excited states. Endeavors to mitigate this quenching effect have typically involved strategies, such as coating an inert/active shell, introducing damping ions to minimize energy migration loss, or reducing surface defects to minimize surface quenching. Furthermore, improving the homogeneous distribution of dopants, using a host with an expanded unit cell dimension or increasing excitation laser irradiance has been shown to relieve cross

[1]MIIT Key Laboratory of Critical Materials Technology for New Energy Conversion and Storage, School of Chemistry and Chemical Engineering, Harbin Institute of Technology, Harbin 150001, PR China. [2]Key Laboratory of Micro-systems and Micro-structures, Ministry of Education, Harbin Institute of Technology, Harbin 150001, PR China. [3]Division of X-ray Photon Science, Department of Physics and Astronomy, Uppsala University, Uppsala 75120, Sweden. ✉ e-mail: chenguanying@hit.edu.cn

relaxation processes and thereby alleviate the concentration quenching effects[21–23]. Despite these progress, the optimal activator concentrations remain rather low, for example, the optimal $Tm^{3+}$ concentration in the $NaYF_4$ host matrix was shown to be within the range of 0.2–1 mol% under 980 nm irradiance of sub-100 $W/cm^2$ [24–26]. Such nanoparticles contain a limited quantity of activator ions, which significantly confines the upconversion brightness.

In this study, we reveal the critical factor of back energy transfer (BET) process from the activator ($Tm^{3+}$) to the sensitizer ($Yb^{3+}$) ions for underlying the lanthanide concentration quenching in the $Yb^{3+}$-$Tm^{3+}$ codoped upconversion systems. We demonstrate that a spatial isolation of $Tm^{3+}$ and $Yb^{3+}$ into the distinct domains of a heterogeneous nanostructure can efficiently suppress the BET process and elevate the optimal $Tm^{3+}$ concentration from 1% to ~8% at sub-100 $W/cm^2$, compared to the typically investigated β-$NaYF_4$:$Yb^{3+}$/$Tm^{3+}$ UCNPs.

## Results

### Heterogeneous core–shell–shell nanostructure design

A heterogeneous core–shell–shell nanostructure was adopted to inhibit both the BET process (from the activator to the sensitizer) and the detrimental surface quenching effect simultaneously (Fig. 1a). In this design, the activator ions (e.g., $Tm^{3+}$) and the sensitizer ions (e.g., $Yb^{3+}$) were incorporated into the core and the first shell domain, respectively. The spatial isolation of the activator and sensitizer ions results in the inhibition of BET processes from the activator ions to the sensitizer ions, while the close contact between the sensitizer and activator ions at the core/shell interface allows efficient energy transfer (ET) across the core–shell boundary. The outermost inert layer is used

to alleviate the surface quenching triggered by migration of excited energy within the first or inner shell layer domain. Specifically, a core–shell–shell nanostructure of $NaYF_4$:x% $Tm^{3+}$/$NaYbF_4$/$NaYF_4$ was devised, in which the inner $NaYbF_4$ shell enriched with $Yb^{3+}$ sensitizer ions was adopted to efficiently capture excitation photons at 980 nm. Subsequently, energy migration among the $Yb^{3+}$ ion network takes places[7], entailing efficient interfacial energy transfers to doped $Tm^{3+}$ ions in the core[27]. Meanwhile, the $NaYF_4$:x% $Tm^{3+}$/$NaYbF_4$ interface could effectively inhibit the BET process ($^3H_4(Tm^{3+}) + {}^2F_{7/2}(Yb^{3+}) \rightarrow {}^3H_6(Tm^{3+}) + {}^2F_{5/2}(Yb^{3+})$). The outermost inert $NaYF_4$ shell can prevent the energy dissipation in the $NaYbF_4$ shell from the quenching effects induced by surface imperfections or external environments[27–29]. This specific nanostructure design facilitates upconversion luminescence (from NIR to VIS/UV) through a ladder-like array of the intermediate metastable states of $Tm^{3+}$ ions, while minimizing nonradiative depopulations of the UCL emitting states (Fig. 2a).

A layer-by-layer epitaxial growth method was used to fabricate a set of the designated core–shell–shell nanoparticles of $NaYF_4$:x% $Tm^{3+}$@$NaYbF_4$@$NaYF_4$ (x = 0.5, 1, 2, 4, 8, 16, 32, 50, 70 and 100) (Fig. 1a). Transmission electron microscopy (TEM) images of the $NaYF_4$:x% $Tm^{3+}$ core, $NaYF_4$:x% $Tm^{3+}$@$NaYbF_4$ core–shell and $NaYF_4$:x% $Tm^{3+}$@$NaYbF_4$@$NaYF_4$ core–shell–shell nanoparticles indicate an average overall size of ~24 nm, with a thickness of ~5.5 nm for the inner layer, and ~3 nm for the outermost shell layer (Supplementary Fig. 1). High resolution TEM (HRTEM, Fig. 1b inset) and selected area electron diffraction patterns (Supplementary Fig. 2b) confirm the single crystalline nature of the designated multilayered nanoparticles. Powder X-ray diffraction (XRD) results confirm that all the resulting core,

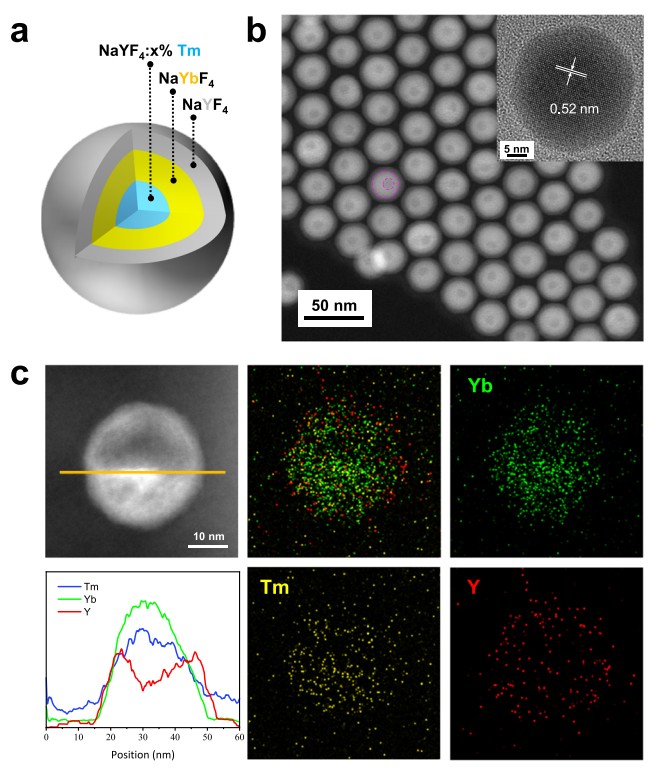

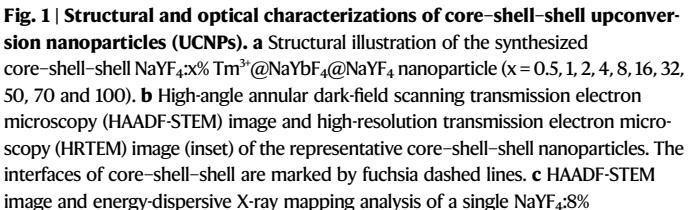

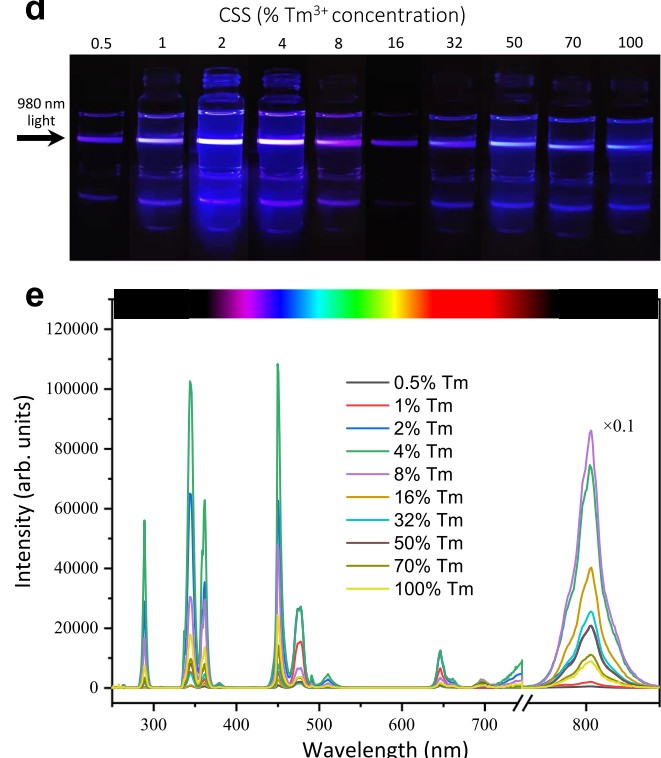

**Fig. 1 | Structural and optical characterizations of core–shell–shell upconversion nanoparticles (UCNPs). a** Structural illustration of the synthesized core–shell–shell $NaYF_4$:x% $Tm^{3+}$@$NaYbF_4$@$NaYF_4$ nanoparticle (x = 0.5, 1, 2, 4, 8, 16, 32, 50, 70 and 100). **b** High-angle annular dark-field scanning transmission electron microscopy (HAADF-STEM) image and high-resolution transmission electron microscopy (HRTEM) image (inset) of the representative core–shell–shell nanoparticles. The interfaces of core–shell–shell are marked by fuchsia dashed lines. **c** HAADF-STEM image and energy-dispersive X-ray mapping analysis of a single $NaYF_4$:8%

$Tm^{3+}$@$NaYbF_4$@$NaYF_4$ nanoparticle, indicating the spatial distribution of the $Yb^{3+}$, $Tm^{3+}$ and $Y^{3+}$ elements in the core–shell–shell structure. **d** Photographic images displaying upconversion luminescence of UCNPs dispersed in n-hexane under 980 nm continuous-wave laser irradiation. CSS, core–shell–shell. **e** Upconversion luminescence spectra acquired under 980 nm excitation at 99.3 $W/cm^2$ for UCNPs in n-hexane. Absorbance of all samples is normalized at 975 nm, corresponding to the $^2F_{7/2} \rightarrow {}^2F_{5/2}$ transition of $Yb^{3+}$ ions. The intensity of 800 nm luminescence peak is multiplied by ×0.1 for better data presentation. Source data are provided as a Source Data file.

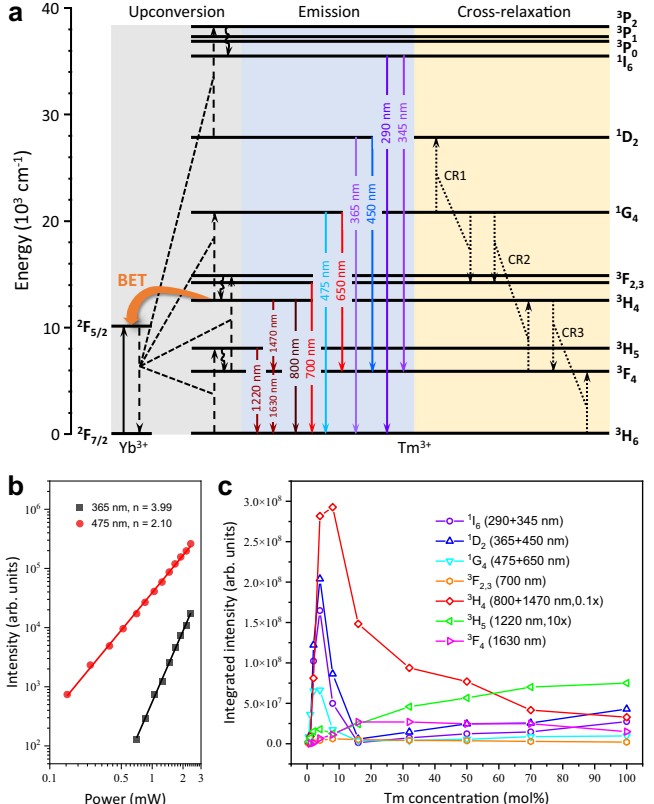

**Fig. 2 | Energy transfer mechanism of Yb³⁺-Tm³⁺ upconversion system under 980 nm excitation. a** Energy level scheme of Yb³⁺ and Tm³⁺ with proposed energy transfer processes under 980 nm laser excitation. Solid line with an arrow, absorption or emission; curve with an arrow, multiphonon relaxation; dashed line with an arrow linked by dotted line, non-radiative energy transfer; BET, back energy transfer. **b** The dependence of 365 and 475 nm upconversion luminescence from NaYF₄:1% Tm³⁺@NaYbF₄@NaYF₄ nanoparticles on excitation laser power in a logarithmic scale. **c** Variation of integrated luminescence intensity of each energy level in Tm³⁺ ions with increasing Tm³⁺ doping concentration (NaYF₄:x% Tm³⁺@NaYbF₄@NaYF₄, x = 0.5, 1, 2, 4, 8, 16, 32, 50, 70, 100) at irradiance of 99.3 W/cm². The luminescence intensity from ³H₄, ³H₅ energy levels is multiplied by ×0.1 and ×10 for better data presentation, respectively. Source data are provided as a Source Data file.

core–shell, and core–shell–shell UCNPs adopt a hexagonal crystal phase (Supplementary Fig. 2a)[30,31]. Atomic-number-contrasted high angle annular dark field scanning transmission electron microscopy (HAADF-STEM) imaging confirms the successful preparation of the heterogeneous core–shell–shell structure (Fig. 1b), with a clear discern of yttrium (Y) atoms (dark) from Yb atoms (bright) that are spatially distributed in the defined way[27,32]. Energy-dispersive X-ray analysis (EDX) verifies the presence of Yb, Tm, Y, Na, F (Supplementary Fig. 3) and the formation of the designated heterogeneous core–shell–shell structure[27,31–33] (Fig. 1c). Inductively-coupled plasma optical emission spectroscopy (ICP-OES) indicates that the actual content of Tm³⁺ (mol %) within the NaYF₄:x% Tm³⁺ core domain is close to the stoichiometric content (Supplementary Table 1).

### Optical characterizations of core–shell–shell upconversion nanoparticles

Representative UCL photographic images of the core–shell–shell UCNPs suspended in hexane and their corresponding UCL spectra are depicted in Fig. 1d, e. The photographic images show both color and luminance change when varying Tm³⁺ concentration in the core of the core–shell–shell nanostructure. At 99.3 W/cm² irradiance of 980 nm laser, we observed that the 8 mol% Tm³⁺ core–shell–shell nanoparticles

emitted an unprecedentedly vibrant upconversion luminescence which markedly surpasses that of 1 mol% Tm³⁺ nanoparticles (the 800 nm luminescence is enhanced approximately 50-fold, Fig. 1e). In contrast, at the same excitation condition, the UCL intensity of the canonical core-only structure NaYF₄:20% Yb³⁺, x% Tm³⁺ (x = 0.2, 0.5, 1, 2, 4, 8, 16) UCNPs initially climbs up and then declines when exceeding 1 mol% Tm³⁺, consistent with previous reports[24–26] (Supplementary Fig. 4). These discoveries suggest that high-efficiency UCL is feasible at considerably high activator concentrations with ~100 W/cm² of 980 nm laser radiation through spatially separating sensitizers and activators into the core and the shell domains. The spatial separation increases the average distance between them and therefore inhibits the BET from activator to sensitizer (Supplementary Fig. 5). This conclusion is supported by the result that co-doping of Tm³⁺ and Yb³⁺ ions into the intermediary shell of NaYF₄@NaYF₄:92% Yb³⁺, 8% Tm³⁺@NaYF₄ nanoparticles will dramatically decreases UCL as compared to the one NaYF₄@NaYF₄:99% Yb³⁺, 1% Tm³⁺@NaYF₄ nanoparticles. Furthermore, the enrichment of Yb³⁺ ions in the intermediary shell of NaYF₄:x% Tm³⁺@NaYbF₄@NaYF₄ nanoparticles favors the intense absorption of the excitation photon energy, thereby enhancing the overall upconversion brightness (Supplementary Fig. 6). Note that the UCL brightness of core–shell–shell NaYF₄:8% Tm³⁺@NaYbF₄@NaYF₄ nanoparticles (~25 nm) is about 4.8-fold higher than that of the state-of-the-art NaYF₄:20% Yb³⁺, 1% Tm³⁺@NaYF₄ core–shell nanoparticles with an overall size of ~ 34 nm (the inert shell thickness, 3.5 nm) (Supplementary Fig. 7). These results validate the rational design of the core–shell–shell nanostructure with spatially separated Tm³⁺ and Yb³⁺ ions. Moreover, the designed core–shell–shell architecture can also be extended to the commonly used Yb³⁺-Er³⁺ and Yb³⁺-Ho³⁺ upconversion systems to accommodate high aviator concentrations for bright upconversion (Supplementary Fig. 8).

### Qualitative understanding of upconversion processes in the Yb³⁺-Tm³⁺ system

We further investigated the dependence of UCL intensity from the high-lying ¹D₂ state of Tm³⁺ on the excitation laser power (Fig. 2b). The number of photons (n) needed to fill up a certain energy level is estimated according to the relationship, $I \propto P^n$, where $P$ is the excitation power and $I$ means the integrated luminescence intensity of interested energy level[34]. For the 475 nm luminescence, $n = 2.10$ was acquired, indicating that three photons are required to excite the ¹G₄ state. While $n = 3.99$ was obtained for the 365 nm UCL, suggesting that populating ¹D₂ state involves a three-photon energy transfer upconversion (ETU) as well as the cross relaxation 1 (CR1) process, as the n value cannot be met purely through ETU processes[35,36].

Moreover, after spectral calibration, the luminescence intensity from each energy level of Tm³⁺ can be compared at different Tm³⁺ concentrations in the designated core–shell–shell nanostructure (Fig. 2c). The highest luminescence intensity derives from the ³H₄ excited state, while the lowest one stems from the ³H₅ state. The luminescence intensities from both energy levels were mathematically scaled appropriately, facilitating the comparison between the different energy levels. As shown in Fig. 2c and Supplementary Table 2, concentration quenching first occurred in the high-lying excited states, including ¹I₆, ¹D₂ and ¹G₄, and then ³H₄, which can be reasonably explained by the existence of the CR2 and CR3 processes between the Tm³⁺ ions[36,37]. Compared with the result of canonical core-only NaYF₄:20% Yb³⁺, x% Tm³⁺ (x = 0.2, 0.5, 1, 2, 4, 8, 16) UCNPs (Supplementary Fig. 9 and Supplementary Table 3), it was confirmed that BET from Tm³⁺ (³H₄) to Yb³⁺ (²F₂/₅) is a dominant factor for the concentration quenching. In stark contrast, a concentration quenching effect was not observed for the 1220 nm down-shifting luminescence originating from the ³H₅ energy level, revealing that ³H₅ does not take part in any pronounced BET or CR processes. Instead, the down-shifting luminescence from the state ³F₄ of Tm³⁺, populated by nonradiative

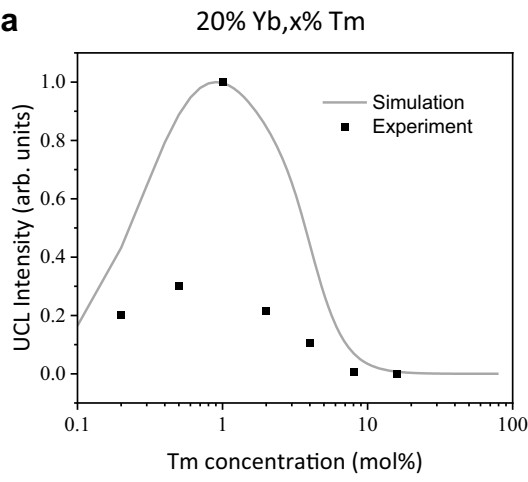

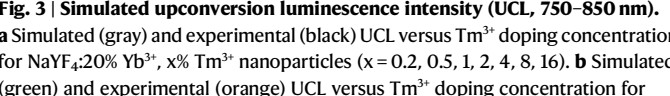

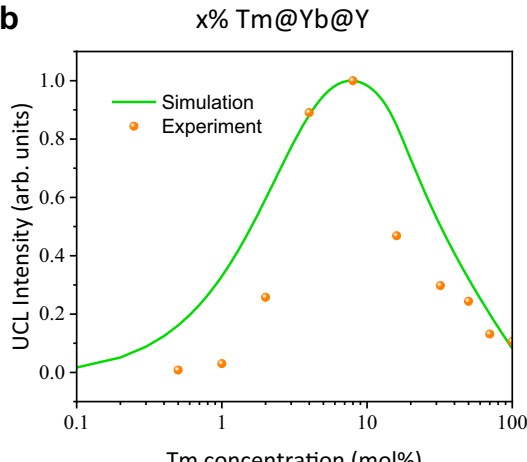

**Fig. 3 | Simulated upconversion luminescence intensity (UCL, 750–850 nm).**
**a** Simulated (gray) and experimental (black) UCL versus $Tm^{3+}$ doping concentration for $NaYF_4$:20% $Yb^{3+}$, x% $Tm^{3+}$ nanoparticles (x = 0.2, 0.5, 1, 2, 4, 8, 16). **b** Simulated (green) and experimental (orange) UCL versus $Tm^{3+}$ doping concentration for

$NaYF_4$:x% $Tm^{3+}$@$NaYbF_4$@$NaYF_4$ nanoparticles (x = 0.5, 1, 2, 4, 8, 16, 32, 50, 70, 100). For all simulations, the 980 nm excitation laser irradiance was set to 99.3 W/cm². Source data are provided as a Source Data file.

relaxation from the $^3H_5$ and radiative relaxations from the $^3H_4$ or other upper excited states, exhibits a complex behavior.

Based on the above outlined steady-state analysis at laser irradiance of 99.3 W/cm², we established a simplified energy level model of the $Tm^{3+}$ ion to simulate the upconversion process in the Yb-Tm system[26,38,39] (Supplementary Fig. 10). Luminescence intensity was calculated as a function of $Tm^{3+}$ doping concentration (range from 0.1 mol% to the max value allowed in each structure). The normalized numerical computation results are in good agreement with the experimental observations for the evolution trend of concentration quenching. Especially, the optimal $Tm^{3+}$ dopant concentration was improved from 1% to 8% (Fig. 3), which further verifies that the designed heterogeneous core–shell–shell nanostructure ($NaYF_4$:x% $Tm^{3+}$@$NaYbF_4$@$NaYF_4$) indeed can weaken the BET from $Tm^{3+}$ ($^3H_4$) to $Yb^{3+}$ ($^2F_{5/2}$), thereby inhibiting the concentration quenching effect.

## Quantitative study of the heterogeneous core–shell–shell nanostructure for inhibiting BET

For quantifying the effect of heterogeneous core–shell–shell nanostructure to inhibit BET, we compared the down-shifting luminescence spectra of $NaYF_4$:1% $Tm^{3+}$@$NaYbF_4$@$NaYF_4$ (6.5@2@3 nm, the number indicates the radius of the core or the thickness of the shell layer) and $NaYF_4$@$NaYbF_4$:1% $Tm^{3+}$@$NaYF_4$ (3.5@5@3 nm) core–shell–shell UCNPs under the same 808 nm excitation condition (Fig. 4a and Supplementary Fig. 11). These two nanostructures have the same overall size, outermost inert protecting shell thickness, as well as the almost equal content ratio and doping concentrations of $Yb^{3+}$/$Tm^{3+}$. The $NaYF_4$@$NaYbF_4$:1% $Tm^{3+}$@$NaYF_4$ nanostructure shows ~2.7-fold peak intensity of 980 nm luminescence (from $Yb^{3+}$ ions) relative to the $NaYF_4$:1% $Tm^{3+}$@$NaYbF_4$@$NaYF_4$ nanostructure, while the intensity of the 1470 nm luminescence stemming from the $^3H_4$ energy level of $Tm^{3+}$ exhibits a contrary tendency. This result gives evidence that the spatial separation of $Tm^{3+}$ and $Yb^{3+}$ in the designated $NaYF_4$:1% $Tm^{3+}$@$NaYbF_4$@$NaYF_4$ heterogenous core–shell–shell structure can efficiently weaken the BET process.

To further support this conclusion and quantify the efficiency of BET process, we produced four pairs of samples, including the first pair $NaYF_4$@$NaYF_4$:1% $Tm^{3+}$@$NaYF_4$ (3.5@5@3 nm) vs. $NaYF_4$@$NaYbF_4$:1% $Tm^{3+}$@$NaYF_4$ (3.5@5@3 nm), the second pair $NaYF_4$:1% $Tm^{3+}$@$NaYF_4$ (3.5@8 nm) vs. $NaYF_4$:1% $Tm^{3+}$@$NaYbF_4$@$NaYF_4$ (3.5@5@3 nm), the third pair $NaYF_4$@$NaYF_4$:8% $Tm^{3+}$@$NaYF_4$ (3.5@5.5@3.5 nm) vs. $NaYF_4$@$NaYbF_4$:8% $Tm^{3+}$@$NaYF_4$ (3.5@5.5@3.5 nm), and the fourth

pair $NaYF_4$:8% $Tm^{3+}$@$NaYF_4$ (3.5@9 nm) vs. $NaYF_4$:8% $Tm^{3+}$@$NaYbF_4$@$NaYF_4$ (3.5@5.5@3.5 nm). We measured the lifetimes of $Tm^{3+}$ at 1470 nm in all the samples under 808 nm pulsed excitation, and then used Eqs. (1) and (2) to quantify the BET efficiencies. The measured lifetimes for each pair are presented in the insets of Fig. 4b, c. The efficiency (rate) of BET process from $Tm^{3+}$ to $Yb^{3+}$ was determined to be 66% (1722 s⁻¹) for the $NaYF_4$@$NaYbF_4$:1% $Tm^{3+}$@$NaYF_4$, and to be 25% (268 s⁻¹) for the $NaYF_4$:1% $Tm^{3+}$@$NaYbF_4$@$NaYF_4$ nanostructure at low $Tm^{3+}$ concentration (Fig. 4b). In analogy, the BET efficiency (rate) was evaluated to be 50% (7077 s⁻¹) for the $NaYF_4$@$NaYbF_4$:8% $Tm^{3+}$@$NaYF_4$, and to be 39% (2095 s⁻¹) for the $NaYF_4$:8% $Tm^{3+}$@$NaYbF_4$@$NaYF_4$ nanostructure at high $Tm^{3+}$ concentration (Fig. 4c). These results clearly demonstrate that the spatial separation of $Tm^{3+}$ activator and $Yb^{3+}$ sensitizer into different domains of the designated core–shell–shell nanostructure can effectively inhibit the BET process. Note that the magnitude of reduction of BET efficiency is smaller at high $Tm^{3+}$ (8%) concentration than at low $Tm^{3+}$ (1%) concentration. This is probably because the cross-relaxation process among $Tm^{3+}$ ions become more severe at high doping concentrations and the reduced mean distance between $Tm^{3+}$ and $Yb^{3+}$ ions aggravates the BET rate. However, the spatial separation of $Tm^{3+}$ and $Yb^{3+}$ ion results in a substantial reduction of BET rate by 4982 s⁻¹ at high $Tm^{3+}$ (8%) concentration, compared to the one of 1454 s⁻¹ at low $Tm^{3+}$ (1%) concentration. Taken together, the designated core–shell–shell nanostructure with spatially separated sensitizer and activator can inhibit the BET process that substantially depopulate the $^3H_4$ state (emitting the most intense 800 nm UCL), thereby alleviating the $Tm^{3+}$ concentration effect. We would like to note that separating $Yb^{3+}$ and $Tm^{3+}$ into different domains simultaneously reduces both ET efficiency and BET efficiency. While the decreased ET efficiency lowers upconversion brightness, the reduced BET efficiency mitigates concentration quenching effects, thereby enhancing brightness. This trade-off in the designed core–shell–shell nanostructure ultimately results in brighter upconversion nanoparticles, as the brightness is more significantly influenced by inhibiting the concentration quenching effect (Supplementary Note).

## Inhibition of activator concentration quenching at high excitation laser irradiance

We then assessed the power-dependent upconversion intensities for the designed core–shell–shell $NaYF_4$:x% $Tm^{3+}$@$NaYbF_4$@$NaYF_4$ (x = 4, 8, 16, 32, 50, 70, 100) nanoparticle at high excitation power densities

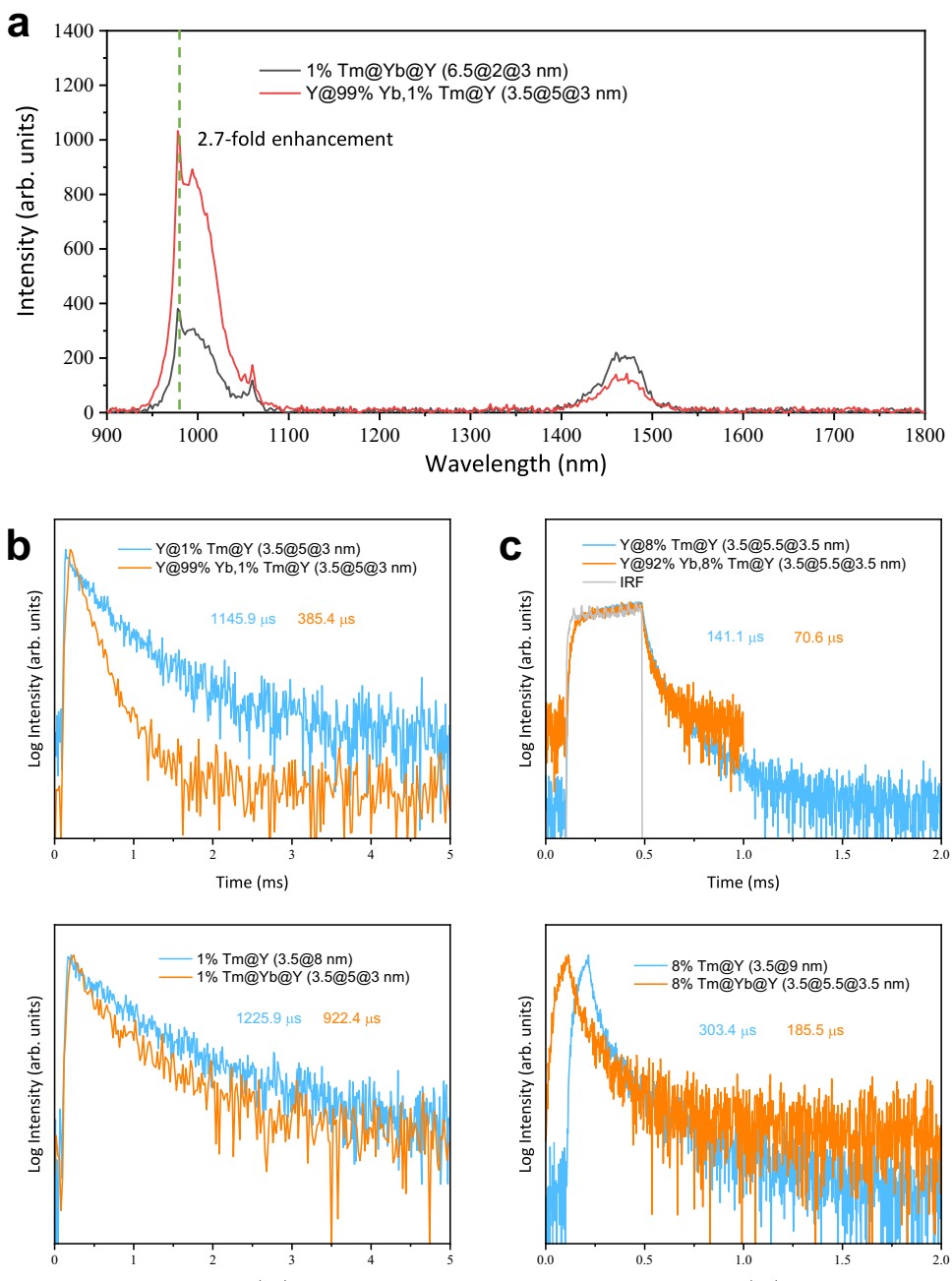

**Fig. 4 | Steady-state and transient measurements to quantify the efficiency of back energy transfer (BET) process from Tm³⁺ to Yb³⁺. a** Down-shifting luminescence spectra under 808 nm excitation at 120 W/cm² for NaYF₄:1% Tm³⁺@NaYbF₄@NaYF₄ and NaYF₄@NaYbF₄:1% Tm³⁺@NaYF₄ upconversion nanoparticles dispersed in n-hexane. Absorbance of the two samples was normalized at 975 nm. The green dashed line indicates the position for comparison of peak intensity. **b, c** Luminescence decay profiles recorded at 1470 nm, corresponding to the $^3H_4 \rightarrow {}^3H_6$ transition of Tm³⁺ ions, under excitation of 808 nm pulsed laser. The calculated average lifetime for each sample is indicated in the figures using color code consistent with the decay curve. IRF denotes the instrument response function. Source data are provided as a Source Data file.

(>100 W/cm²) (Fig. 5a) utilizing a home-built confocal microscope (Supplementary Fig. 12). Increasing 980 nm laser irradiance from 100 W/cm² to 20 MW/cm² enhances the overall (400–850 nm) UCL intensity, accompanied with a gradually decreasing slope that finally reaches a plateau for all samples. The relative magnitude of UCL intensity at 100 W/cm² for the samples of 4% and 8% Tm³⁺ is consistent with the ensemble measurement results in Fig. 2c.

We further quantified and compared the average upconversion brightness of single UCNPs for all the heterogenous core–shell–shell samples at 20 MW/cm² (Fig. 5b). The detailed histograms of single particle brightness for each sample are shown in Supplementary Fig. 13. For 0.5, 1 and 2% Tm³⁺ doped samples, which were not bright enough to surpass the detection limit, we estimated their average brightness according to the inhibiting effect of concentration quenching for other samples. No noticeable UCL photo-bleaching or photo-blinking was observed over 60 mins single nanoparticle measurements, affirming the stable structure of the designated core–shell–shell UCNPs at high laser irradiances (Supplementary Fig. 14). The overall upconversion brightness (UCL, 400–850 nm) shows a nearly linear growth with the increment of Tm³⁺ concentration

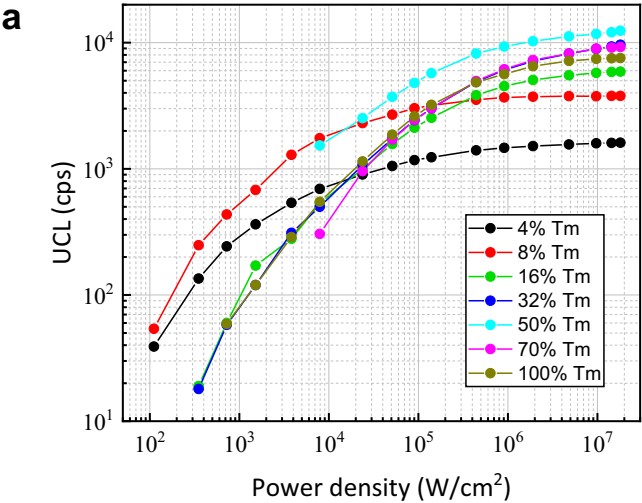

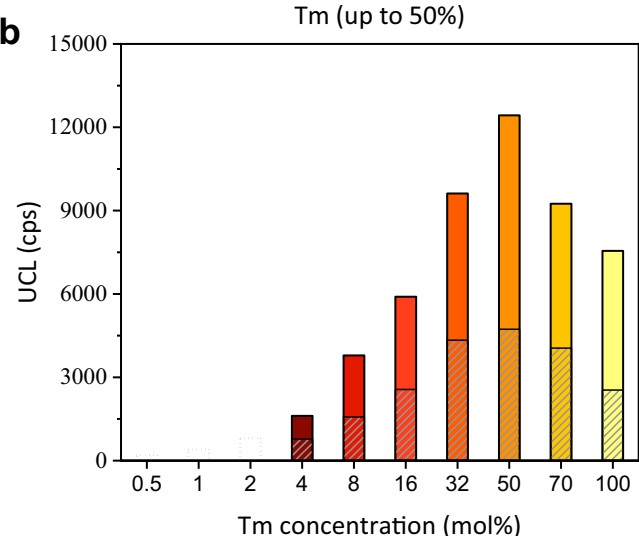

**Fig. 5 | Analysis on upconversion efficiency influenced by power. a** Consolidated UCL (400–850 nm) log-log plotted with 980 nm laser irradiance for the core–shell–shell NaYF$_4$:x% Tm$^{3+}$@NaYbF$_4$@NaYF$_4$ (x = 4, 8, 16, 32, 50, 70 and 100) UCNPs. **b** Average single particle UCL brightness for all the core–shell–shell samples under laser irradiance of 20 MW/cm$^2$; note that UCL from the core–shell–shell samples with 0.5, 1, and 2% Tm$^{3+}$ is marked by dotted box, as these samples were not bright enough to be measured. The 800 nm luminescence (760-850 nm) intensity is marked with a gray stripe pattern. Luminescence intensity at 20 MW/cm$^2$ in (**b**) was determined by single particle measurements, while the variation of each saturation curve in (**a**) was given by multi-particle investigations[31]. Source data are provided as a Source Data file.

up to 50%, which is about six-fold higher than the optimal Tm$^{3+}$ concentration of 8% at laser irradiance of 99.3 W/cm$^2$. The brightness of the UCL band at 800 nm exhibits a similar trend as the overall upconversion brightness with an optimal Tm$^{3+}$ doping concentration of 50%. Note that the intensities of shorter wavelength UCL (400−760 nm) from higher-lying energy levels ($^1G_4$, $^1D_2$, $^1I_6$) (three-, four- and five-photon processes) are on par with that of UCL at 800 nm (two-photon process) under single UCNP measurement. This can be attributed to the involvement of higher-order multiphoton processes and the utilization of high laser irradiances. Importantly, the high laser irradiance induced increase of the optimized Tm$^{3+}$ concentration affirms the existence of the contribution of the cross-relaxation processes to the activator concentration quenching, alongside the revealed BET process from Tm$^{3+}$ ($^3H_4$) to Yb$^{3+}$ ($^2F_{5/2}$) at low laser

irradiances. In other words, both the BET processes and the cross-relaxation processes collectively define the activator Tm$^{3+}$ concentration quenching effect in UCNPs.

## Discussion

In this work, we highlight the crucial role of back energy transfer (BET) from activator Tm$^{3+}$ to sensitizer Yb$^{3+}$ in the lanthanide concentration quenching of Yb$^{3+}$/Tm$^{3+}$-codoped UCNPs. Through the strategic design and synthesis of the heterogenous core−shell−shell structure, we demonstrated that segregating sensitizer and activator ions into adjacent layers effectively suppresses BET while entailing efficient interfacial energy transfer from Yb$^{3+}$ to Tm$^{3+}$, resulting in bright upconversion at high Tm$^{3+}$ doping concentrations. As a consequence, the optimal Tm$^{3+}$ concentration increased to ~8%, compared with the typically investigated β-NaYF$_4$:Yb$^{3+}$/Tm$^{3+}$ UCNPs, under excitation power densities below 100 W/cm$^2$, which is in good agreement with rate equation modeling. At higher excitation power densities of about 20 MW/cm$^2$, the optimal Tm$^{3+}$ concentration reached an unprecedented ~50% at the single particle level. Moreover, minimizing lattice defects to reduce excited energy losses may further enhance the optimal Tm$^{3+}$ doping concentration[32]. We believe this segregated doping strategy will advance the design of bright UCNPs for a variety of applications.

## Methods
### Nanoparticle synthesis
NaYF$_4$:x% Tm$^{3+}$@NaYbF$_4$@NaYF$_4$ and other core-multishell nanoparticles were synthesized following a seed-mediated hot-injection procedure. The NaYF$_4$:x% Tm$^{3+}$ core is first synthesized to serve as a template seed for layer-by-layer epitaxial growth of the two shells in sequence. While the core-only structure NaYF$_4$:20% Yb$^{3+}$, x% Tm$^{3+}$ was prepared using a coprecipitation protocol. Experimental details are described in Supplementary Methods.

### Ensemble characterization
The size, shape and spatial distribution of elements were examined using a transmission electron microscope (FEI Tecnai G2 F30 S-TWIN) hyphenated energy dispersive X-ray spectroscopy operating at 300 kV accelerating voltage, while the phase and crystalline structure of samples were analyzed by a powder X-ray diffractometer (BRUKER D8 ADVANCE). Tm$^{3+}$ molar content was determined with an inductively-coupled plasma optical emission spectrometer (Thermo Fisher Scientific ICAP 6000). Steady-state photoluminescence and transient decay behavior were measured by a spectrofluorometer (FLS1000, Edinburgh) integrated with 808 nm (MDL-III-808-2W, CNI) and 980 nm (MDL-III-980-2W, CNI) diode lasers at room temperature.

### Calculation of BET efficiency
The back energy transfer (from Tm$^{3+}$ to Yb$^{3+}$) efficiency η can be quantitatively estimated on the basis of Eqs. (1) and (2)[40]:

$$\eta = 1 - \frac{\tau_m}{\tau_{in}} \quad (1)$$

$$\tau_m = \frac{\sum \alpha_i \tau_i^2}{\sum \alpha_i \tau_i} \quad (2)$$

where $\tau_m$ is the mean lifetime of energy donor (Tm$^{3+}$) when in the company of energy acceptor (Yb$^{3+}$), $\tau_{in}$ is the intrinsic lifetime of energy donor (Tm$^{3+}$) determined by replacing energy acceptor (Yb$^{3+}$) with inert Y$^{3+}$, and α is the amplitude.

### Single particle measurements
The luminescence intensity of a single nanoparticle was measured by a self-built confocal microscope. Supplementary Fig. 12 draws the

schematic plot of experimental apparatus, wherein UCNPs are excited by a single-mode fibre coupled 980 nm diode laser (VLSS-980-B, Connet). The purpose of the first polarizer (DGL10, Thorlabs) is to produce a polarized laser beam. Based on this, a second polarizer (GTH10M, Thorlabs) is employed to manager the excitation power, facilitated with rotation of a half-wave plate (H, WPH10E-980, Thorlabs). The whole excitation and detection process: the collimated excitation beam is reflected by a short pass dichroic mirror (DM, ZT1064rdc-sp-UF1, Chroma), and focused through an air objective lens (UPlanSApo, 40×/0.95, Olympus) onto the sample slide. Photoluminescence is collected via the same objective and then split from the excitation beam by the DM. After filtered by a short pass filter (SPF, FF01-950SP-25, Semrock), the luminescence signal is coupled into a 4f-system containing a 100 μm diameter pinhole (P100K, Thorlabs), and detected by a photomultiplier tube (PMT, HPM-100-42, Becker & Hickl GmbH).

**Theoretical modeling and numerical simulation**

The simplified energy level diagram of the $Tm^{3+}$ ion is illustrated in Supplementary Fig. 10. Upconversion process in a Yb-Tm system can be depicted by the following rate equations:

$$Tm^{3+}\ (^3H_6):\ \frac{d}{dt}n_{A_0} = -\sum_{i=1}^{4}\frac{d}{dt}n_{A_i} \tag{3}$$

$$Tm^{3+}\ (^3F_4):\ \frac{d}{dt}n_{A_1} = \beta_2 n_{A_2} + 2c_1 n_{A_0} n_{A_3} - w_1 n_{S_1} n_{A_1} - \frac{n_{A_1}}{\tau_{A_1}} \tag{4}$$

$$Tm^{3+}\ (^3H_5):\ \frac{d}{dt}n_{A_2} = w_0 n_{S_1} n_{A_0} - \beta_2 n_{A_2} - \frac{n_{A_2}}{\tau_{A_2}} \tag{5}$$

$$Tm^{3+}\ (^3H_4):\ \frac{d}{dt}n_{A_3} = \beta_4 n_{A_4} - (\Gamma_r^A + c_1 n_{A_0} + w_b n_{S_0})n_{A_3} \tag{6}$$

$$Tm^{3+}\ (^3F_{2,3}):\ \frac{d}{dt}n_{A_4} = w_1 n_{S_1} n_{A_1} - \beta_4 n_{A_4} - \frac{n_{A_4}}{\tau_{A_4}} \tag{7}$$

$$Yb^{3+}\ (^2F_{7/2}):\ \frac{d}{dt}n_{S_0} = -\frac{d}{dt}n_{S_1} \tag{8}$$

$$Yb^{3+}\ (^2F_{5/2}):\ \frac{d}{dt}n_{S_1} = \frac{\sigma_S I}{h\nu}n_{S_0} - \Gamma_r^S n_{S_1} - (w_0 n_{A_0} + w_1 n_{A_1})n_{S_1} + w_b n_{S_0} n_{A_3} \tag{9}$$

Here $n_{A_i}$ (i = 0–4) and $\tau_{A_i}$ (i = 1–4; $\tau_{A_3} = 1/\Gamma_r^A$) represent the population density and radiative lifetime of $^3H_6$, $^3F_4$, $^3H_5$, $^3H_4$ and $^3F_{2,3}$ energy levels for $Tm^{3+}$, respectively. $w_i$ (i = 0, 1) is the energy transfer rate from the $^2F_{5/2}$ state of $Yb^{3+}$ to the $^3H_5$ and $^3F_{2,3}$ states of $Tm^{3+}$, respectively. $w_b$ is the back energy transfer rate from the $^3H_4$ state of $Tm^{3+}$ to the $^2F_{5/2}$ state of $Yb^{3+}$. $c_1$ denotes the rate of cross relaxation for $Tm^{3+}$-$Tm^{3+}$ ($^3H_4 + {}^3H_6 \rightarrow {}^3F_4 + {}^3F_4$). $\beta_i$ (i = 2, 4) is the nonradiative decay rate for $^3H_5$ and $^3F_{2,3}$ energy levels of $Tm^{3+}$, respectively. $n_{S_0}$ and $n_{S_1}$ represent the population density of ground state and excited state for $Yb^{3+}$, respectively. $\tau_{S_1}(1/\Gamma_r^S)$ is the lifetime of excited state for $Yb^{3+}$. $\sigma_S$ is the absorption cross section of $Yb^{3+}$ at 980 nm. $h$ is Planck constant. $\nu$ is the frequency of 980 nm laser light. $I$ is the excitation irradiance of 980 nm laser beam.

For NaYF$_4$:x% Tm@NaYbF$_4$@NaYF$_4$ core-multishell system, $Yb^{3+}$ and $Tm^{3+}$ are equivalently regarded to uniformly distribute in the active region (NaYF$_4$:x% $Tm^{3+}$@NaYbF$_4$) as a co-doping condition, while holding the amount of ions constant. Nonetheless, cross relaxation of $Tm^{3+}$-$Tm^{3+}$ was excluded from this assumption. In addition, the influence of varying Tm concentration upon the parameters $w_0$, $w_1$, $c_1$ related to energy transfer between $Yb^{3+}$ and $Tm^{3+}$ is referred to the description provided by Lee et al.[39]. The rate parameters for core-only (NaYF$_4$:20% $Yb^{3+}$, 1% $Tm^{3+}$) and core–shell–shell (NaYF$_4$:1% $Tm^{3+}$@NaYbF$_4$@NaYF$_4$) samples are listed in Supplementary Table 4.

**Reporting summary**

Further information on research design is available in the Nature Portfolio Reporting Summary linked to this article.

## Data availability

All the relevant data that support the findings of this work are available from the corresponding author upon request. Source data are provided with this paper.

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

## Acknowledgements

This work was supported by the Fundamental Research Fund for Distinguished Scholars (Harbin Institute of Technology) (XWQQ5710001615 to G.C.), the Fundamental Research Funds for the Central Universities, China (AUGA5710052614 and AUGA8880100415 to G.C.), and the National Natural Science Foundation of China (52272270 and 51972084 to G.C.).

## Author contributions

G.C. conceived the research; D.H. prepared nanoparticles under the guidance of F.L.; D.H. built the optical system; D.H. acquired, processed and analyzed data; D.H. developed the theoretical model and performed numerical simulations; The paper was written by D.H. and G.C., with comments from H.Å.; All authors discussed the results; The project was supervised by G.C.

## Funding

## Competing interests

The authors declare no competing interests.
