## [Transparent Peer Review file · Nature Communications]

Inhibiting Concentration Quenching in Yb³⁺-Tm³⁺ Upconversion Nanoparticles by Suppressing Back Energy Transfer

Corresponding Author: Professor Guanying Chen

Version 0:

Reviewer comments:

Reviewer #1

(Remarks to the Author)

In the manuscript entitled "Breaking Concentration Quenching of Yb³⁺-Tm³⁺ Upconversion Systems by Suppressing Back Energy Transfer", Huang et al. reported the use of NaYF₄:Tm@NaYbF₄@NaYF₄ nanoparticles for breaking the concentration quenching. However, the referee believes that the threshold of requirements for publication of this manuscript in Nature Communications is not met due to the following queries:

1. The concepts of "core-shell structure", "back energy transfer", and "heavy lanthanide doping" are not new for upconversion systems. The use of high-power excitation to reduce cross-relaxation for UCNPs is also routine work. Much work has been done to investigate the concentration effect and high-power effect. I think this work does not show an outstanding contribution to support its publication in such a high-standard journal.
2. I would like to say that due to the inadequate energy transfer (because of the spatially separated Tm and Yb), the effective doping concentration may be much lower than the nominal doping concentration. Therefore, I can not accept the saying "breaking the concentration quenching" since the authors do not have further evidence.
3. The authors utilized a rate equation to simulate the energy transfer process for both NaYF₄:Yb,Tm@NaYF₄, and NaYF₄:Tm@NaYbF₄@NaYF₄ nanoparticles. But seeing the data from Figure S6 and Table S3 (especially from w₀, w₁, and w₂), it is obvious that by increasing the average distance between Yb and Tm at different layers, both Yb-Tm energy transfer and Tm-Yb BET are suppressed. That is to say, the breaking of the concentration quenching is at the cost of losing ET efficiency. Also, it is still unclear the emission performance between the optimal NaYF₄:Tm@NaYbF₄@NaYF₄ nanoparticle and the state-of-the-art NaYF₄:Yb,Tm@NaYF₄ nanoparticles. Why did the authors choose these numerical values of w₀ (1.7×10⁻¹⁷ vs 7.7×10⁻¹⁸), w₁ (3.5×10⁻¹⁶ vs 1.5×10⁻¹⁶), w₂ (1.7×10⁻¹⁸ vs 7.7×10⁻¹⁹), is also unclear.

Reviewer #2

(Remarks to the Author)

The effect of lanthanide concentration quenching has long been known as a critical issue that limits the upconversion luminescence brightness of lanthanide-doped materials.

In this manuscript, the authors demonstrate the inhibition of backward energy transfer from activator(Tm) to sensitizer(Yb) ions, utilizing a core-shell-shell structure, can significantly suppress this effect and increase the optimized Tm concentration from 0.5% to 8%.

The conclusions look convincing and reasonably supported by properly designed experiments and quantitative results.

Therefore, I think this manuscript is suitable for publication in Nature Communications after reflecting minor revisions as indicated below.

- 1) Supplementary Figure 2a: there are two small sharp peaks located on the pattern of the 8% Tm core only (at about 38° and 56° of 2θ) which are not matching the standard ones of β-NaYF₄. Please clarify this.
- 2) Figure 2c: it seems that the optimized doping concentration is lower for higher-lying emission levels. Please explain this in detail. Also as a minor one, please replace "Integrated Intensity" with "Integrated intensity".

- 3) Figure 5b: please consider analysis for different emission band.
- 4) Supplementary Figure 4: please add the corresponding size distribution of samples in (a)-(g).
- 5) Supplementary Figure 8: please mark the size of each layer for all samples.

Reviewer #3

(Remarks to the Author)

This study focuses on enhancing the upconversion efficiency of lanthanide-doped NaYF₄-based nanoparticles, specifically by addressing the concentration quenching effect that limits their brightness under low excitation irradiance (<100 W/cm²). The authors designed a heterogeneous core-shell-shell structure in which Tm³⁺ activator ions are confined to the core, while Yb³⁺ sensitizer ions are distributed in the inner shell. This design minimizes the back energy transfer (BET) from Tm³⁺ to Yb³⁺, a significant factor contributing to concentration quenching, and enables an increase in optimal Tm³⁺ doping concentration from 0.5% to 8%. At higher excitation irradiance (~20 MW/cm²), the optimal Tm³⁺ concentration can be pushed to as high as 50%, which is unprecedented for single-particle systems. This work provides new insights into enhancing upconversion nanoparticle (UCNP) efficiency and lays the groundwork for broader applications in fields like biosensing, bioimaging, and solid-state lasing. Therefore, I recommend it for publication after addressing the minor points:

1. While the study focuses on Yb-Tm systems, it would be beneficial to discuss whether the same architecture could be applied to other lanthanide pairs, such as Yb-Er or Yb-Ho, which are also commonly used in upconversion applications.
2. The long-term stability of the core-shell-shell structure under extended irradiation conditions was not discussed in detail. Information on the stability of the designed architecture would further enhance the impact of the study.

Version 1:

Reviewer comments:

Reviewer #1

(Remarks to the Author)

Chen et al. submitted a revised article related to Inhibiting Concentration Quenching of Yb³⁺-Tm³⁺ Upconversion Systems by Suppressing Back Energy Transfer. The current version of the article has addressed the concerns of the reviewers.

Reviewer #2

(Remarks to the Author)

In general, I am quite satisfied by the author's revision this time. I have printed all they submitted for reviewers including main manuscript, and read those line by line. I think now, since this revised version have solved most of the issues raised by reviewers, it can be published in its current form.

Reviewer #3

(Remarks to the Author)

The previous reviewers have provided numerous constructive comments, and the authors have made revisions and additions in response to these suggestions. The current version appears to be more publishable. Therefore, I recommend an acceptance of the manuscript.

Manuscript No. NCOMMS-24-59146

Title: Breaking Concentration Quenching of Yb^{3+} - Tm^{3+} Upconversion Systems by Suppressing Back Energy Transfer

Revised Title: **Inhibiting** Concentration Quenching of Yb^{3+} - Tm^{3+} Upconversion Systems by Suppressing Back Energy Transfer

Firstly, we thank all reviewers very much for their precious comments on our manuscript, which helped us significantly improve the manuscript. In the following, we provide a point-by-point response to the reviewers' comments. The reviewers' comments are written in **blue**, while our responses to them are written in **black**. The revisions in the manuscript are indicated in **red**.

Response to reviewers' comments

Reviewer #1 (Remarks to the Author):

In the manuscript entitled “Breaking Concentration Quenching of Yb^{3+} - Tm^{3+} Upconversion Systems by Suppressing Back Energy Transfer”, Huang et al. reported the use of $\text{NaYF}_4:\text{Tm}@\text{NaYbF}_4@\text{NaYF}_4$ nanoparticles for breaking the concentration quenching. However, the referee believes that the threshold of requirements for publication of this manuscript in Nature Communications is not met due to the following queries:

Response: We thank for the reviewer's comments helping us to significantly improve the paper.

1. The concepts of “core-shell structure”, “back energy transfer”, and “heavy lanthanide doping” are not new for upconversion systems. The use of high-power excitation to reduce cross-relaxation for UCNPs is also routine work. Much work has been done to investigate the concentration effect and high-power effect. I think this work does not show an outstanding contribution to support its publication in such a high-standard journal.

Response: The novelty of this work does not refer to a first-time proposal of any of the concepts “core-shell structure”, “back energy transfer” or “heavy lanthanide doping” for upconversion systems. Rather, the novelty of this work relates to the development of a core-shell upconversion system that substantially weakens the back energy transfer, which here is revealed to be the main factor for concentration quenching of activator ions in such systems. In particular, we construct a heterogeneous core-shell-shell structure $\text{NaYF}_4:x\% \text{Tm}^{3+}@\text{NaYbF}_4@\text{NaYF}_4$, in which activator (Tm^{3+}) and sensitizer (Yb^{3+}) are separated at different regions to weaken the back energy transfer from Tm^{3+} to Yb^{3+} , resulting in an

increase of optimal Tm^{3+} doping concentration from 1% to 8% at sub-100 W/cm^2 irradiance of 980 nm laser, compared with the typically investigated $\beta\text{-NaYF}_4\text{:Yb}^{3+}/\text{Tm}^{3+}$ upconversion nanoparticles. This groundbreaking advancement infers novelty.

Moreover, combined with high excitation laser irradiance ($20 \text{ MW}/\text{cm}^2$), the optimal Tm^{3+} doping concentration can reach 50%, which is unprecedented for single particle systems. This breakthrough verifies that cross-relaxation is indeed another critical factor of concentration quenching, and more importantly reconfirms that back energy transfer is the significant factor contributing to lanthanide concentration quenching regardless of the excitation conditions, compared to the published work (ref. 26, *Nat. Nanotechnol.* 2013, 8, 729-734; the main result in ref. 26 is that the optimal Tm^{3+} doping concentration can rise from 0.5% to 8% in the core-only structure $\text{NaYF}_4\text{:20% Yb}^{3+}, x\% \text{Tm}^{3+}$ when increasing laser irradiance from $10 \text{ W}/\text{cm}^2$ to $2.5 \text{ MW}/\text{cm}^2$).

Above all, through a quantitative investigation, we comprehensively demonstrated that back energy transfer and cross-relaxation are the primary mechanisms underlying concentration quenching when surface quenching is effectively mitigated. Furthermore, we established that our designed heterogeneous core-shell-shell structure provides an effective strategy to suppress back energy transfer, which, when combined with high excitation laser irradiance, offers a near-complete solution to the longstanding issue of concentration quenching. We believe that this easily implemented segregated doping strategy and mechanistic insights provided by this work will markedly promote the understanding of the photophysical properties of lanthanide UCNPs and substantially advance the design rationale of bright lanthanide upconversion materials for the benefit of diverse applications.

2. I would like to say that due to the inadequate energy transfer (because of the spatially separated Tm and Yb), the effective doping concentration may be much lower than the nominal doping concentration. Therefore, I can not accept the saying “breaking the concentration quenching” since the authors do not have further evidence.

Response: We supplemented ICP-OES measurements to determine the actual doping concentration of the core samples $\text{NaYF}_4\text{:}x\% \text{Tm}^{3+}$ ($x = 0.5, 1, 2, 4, 8, 16, 32, 50, 70$) (Table R1). The results indicated that the actual doping concentrations of all measured samples are all close to the nominal doping concentrations. In addition, to address the reviewer’s concern on the saying “breaking the concentration quenching”, we have revised the title into “**Inhibiting** Concentration Quenching of $\text{Yb}^{3+}\text{-Tm}^{3+}$ Upconversion Systems by

Suppressing Back Energy Transfer”.

Table R1. The detailed Tm^{3+} doping concentration in $\text{NaYF}_4:x\% \text{Tm}^{3+}$ core nanoparticles determined by inductively-coupled plasma optical emission spectroscopy (ICP-OES) analysis.

$\text{NaYF}_4:x\% \text{Tm}^{3+}$ (nominal mol%)	ICP-OES measurement (mol%)
x = 0.5	0.3
x = 1	0.7
x = 2	1.5
x = 4	3.9
x = 8	8.1
x = 16	13.7
x = 32	27.4
x = 50	47.7
x = 70	65.6

Action. We have included Table R1 as Supplementary Table 1 in the revised SI, and made the following corresponding changes to the revised manuscript.

Page 3, paragraph 1. “Inductively-coupled plasma optical emission spectroscopy (ICP-OES) indicates that the actual content of Tm^{3+} (mol%) within the $\text{NaYF}_4:x\% \text{Tm}^{3+}$ core domain is close to the stoichiometric content (**Supplementary Table 1**)”

Revised Title. “Inhibiting Concentration Quenching of Yb^{3+} - Tm^{3+} Upconversion Systems by Suppressing Back Energy Transfer”.

3. The authors utilized a rate equation to simulate the energy transfer process for both $\text{NaYF}_4:\text{Yb},\text{Tm}@ \text{NaYF}_4$, and $\text{NaYF}_4:\text{Tm}@ \text{NaYbF}_4@ \text{NaYF}_4$ nanoparticles. But seeing the data from Figure S6 and Table S3 (especially from w_0 , w_1 , and w_2), it is obvious that by increasing the average distance between Yb and Tm at different layers, both Yb-Tm energy transfer and Tm-Yb BET are suppressed. That is to say, the breaking of the concentration quenching is at the cost of losing ET efficiency. Also, it is still unclear the emission performance between the optimal $\text{NaYF}_4:\text{Tm}@ \text{NaYbF}_4@ \text{NaYF}_4$ nanoparticle and the

state-of-the-art NaYF₄:Yb,Tm@NaYF₄ nanoparticles. Why did the authors choose these numerical values of w_0 (1.7×10^{-17} vs 7.7×10^{-18}), w_1 (3.5×10^{-16} vs 1.5×10^{-16}), w_b (1.7×10^{-18} vs 7.7×10^{-19}), is also unclear.

Response: We appreciate the reviewer's comment. Indeed, separating Yb³⁺ and Tm³⁺ into different domains simultaneously reduces both energy transfer (ET) efficiency and back energy transfer (BET) efficiency. While the decreased ET efficiency lowers upconversion brightness, the reduced BET efficiency mitigates concentration quenching effects, thereby enhancing brightness. This trade-off in the designed core-shell-shell nanostructure ultimately results in brighter upconversion nanoparticles, as the brightness is more significantly influenced by inhibiting the concentration quenching effect.

To validate this conclusion, we synthesized a series of NaYF₄:x% Yb³⁺, 4% Tm³⁺@NaYbF₄@NaYF₄ (3.5@5@3 nm) (x=0, 5, 10, 20) as control samples (Figure R1). Varying doping levels of Yb³⁺ ions in the core domain will simultaneously enhances both the ET efficiency from Yb³⁺ to Tm³⁺ and also the BET efficiency from Tm³⁺ to Yb³⁺. If the ET process dominates over BET in influencing upconversion, an increase in Yb³⁺ concentration should enhance upconversion luminescence (UCL) brightness; otherwise, it would lead to a decrease. As shown in Figure R1, increasing the Yb³⁺ doping levels in the core domain consistently reduces the UCL brightness, indicating that BET has a more pronounced effect than ET on upconversion brightness.

Figure R1. Upconversion luminescence spectra collected under 980 nm laser irradiance at 99.3 W/cm² for NaYF₄:x% Yb³⁺, 4% Tm³⁺@NaYbF₄@NaYF₄ (3.5@5@3 nm) (x=0, 5, 10, 20) nanoparticles dispersed in n-hexane. The absorbance of all samples has been normalized at 975 nm for the ²F_{7/2} →

$^2F_{5/2}$ transition of Yb^{3+} ions.

To further support this conclusion, we subsequently prepared a series of $NaYF_4:20\% Yb^{3+}, x\% Tm^{3+}@NaYbF_4@NaYF_4 (3.5@5@3\text{ nm})$ ($x=1, 2, 4, 8$) control samples, in which the core domain contains additional 20 mol% Yb^{3+} dopants to enhance energy transfer from Yb^{3+} to Tm^{3+} while varying the concentrations of Tm^{3+} (Figure R2). First, the optimal Tm^{3+} concentration was determined to be 2 mol%, in contrast to 8% for $NaYF_4:x\% Tm^{3+}@NaYbF_4@NaYF_4 (3.5@5@3\text{ nm})$ ($x=1, 2, 4, 8$), respectively. This finding reconfirms the importance of spatially isolating Yb^{3+} and Tm^{3+} for inhibiting the concentration quenching effect. Second, at low Tm^{3+} dopant concentrations ($x=1$ and 2), $NaYF_4:20\% Yb^{3+}, x\% Tm^{3+}@NaYbF_4@NaYF_4 (3.5@5@3\text{ nm})$ exhibits higher UCL than $NaYF_4:x\% Tm^{3+}@NaYbF_4@NaYF_4$ samples. At higher Tm^{3+} concentrations ($x=4$ and 8), the brightness decreases, making these samples dimmer than that of $NaYF_4:x\% Tm^{3+}@NaYbF_4@NaYF_4$. This suggests that at low Tm^{3+} concentrations, ET dominates over BET in influencing UCL brightness, whereas at high Tm^{3+} concentrations, BET plays a more significant role than ET. Among all samples, the brightest one was determined to be $NaYF_4:8\% Tm^{3+}@NaYbF_4@NaYF_4$, highlighting the advantage of spatially separated Yb^{3+} and Tm^{3+} ions in the designated core–shell–shell nanostructure.

Figure R2. (a) Upconversion luminescence (UCL) spectra collected under 980 nm excitation at 99.3 W/cm² for $NaYF_4:20\% Yb^{3+}, x\% Tm^{3+}@NaYbF_4@NaYF_4 (3.5@5@3\text{ nm})$ ($x=1, 2, 4, 8$) nanoparticles dispersed in n-hexane. The absorbance of all samples has been normalized at 975 nm for the $^2F_{7/2} \rightarrow ^2F_{5/2}$ transition of Yb^{3+} ions. (b) UCL intensity (250–850 nm) for $NaYF_4:20\% Yb^{3+}, x\% Tm^{3+}@NaYbF_4@NaYF_4 (3.5@5@3\text{ nm})$ and $NaYF_4:x\% Tm^{3+}@NaYbF_4@NaYF_4 (3.5@5@3\text{ nm})$ ($x=1, 2, 4, 8$) under 980 nm laser irradiance of 99.3 W/cm². The ratio of UCL brightness for $NaYF_4:20\% Yb^{3+}, x\% Tm^{3+}@NaYbF_4@NaYF_4$ to that for $NaYF_4:x\% Tm^{3+}@NaYbF_4@NaYF_4$ at each Tm^{3+} concentration is marked on the top of blue column for corresponding $NaYF_4:20\% Yb^{3+}, x\% Tm^{3+}@NaYbF_4@NaYF_4$ sample.

Per the suggestion of the reviewer, we also compared the UCL intensity of NaYF₄:8% Tm³⁺@NaYbF₄@NaYF₄ (3.5@5.5@3.5 nm) core-shell-shell nanoparticles with that of the state-of-the-art NaYF₄:20% Yb³⁺, 1% Tm³⁺@NaYF₄ (13.5@3.5 nm) core-shell nanoparticles (Figure R3) under identical measurement conditions. Despite having a smaller size, NaYF₄:8% Tm³⁺@NaYbF₄@NaYF₄ nanoparticles (~ 25 nm) have higher upconversion efficiency than the state-of-the-art NaYF₄:20% Yb³⁺, 1% Tm³⁺@NaYF₄ (34 nm) nanoparticles with a larger size. For UCL brightness per unit volume (Counts/nm³), our designed NaYF₄:8% Tm³⁺@NaYbF₄@NaYF₄ nanoparticles exhibit fivefold higher UCL brightness than the state-of-the-art NaYF₄:20% Yb³⁺, 1% Tm³⁺@NaYF₄ core-shell nanoparticles.

Figure R3. TEM image (**a**) and size distribution (**b**) of state-of-the-art NaYF₄:20% Yb³⁺, 1% Tm³⁺@NaYF₄ (13.5@3.5 nm) core-shell nanoparticles. The size distribution is fitted by a Gaussian curve (red line). (**c**) Upconversion luminescence spectra for NaYF₄:20% Yb³⁺, 1% Tm³⁺@NaYF₄ (13.5@3.5 nm) and NaYF₄:8% Tm³⁺@NaYbF₄@NaYF₄ (3.5@5.5@3.5 nm) nanoparticles dispersed in n-hexane. The absorbance of all samples has been normalized at 975 nm for the ²F_{7/2} → ²F_{5/2} transition of Yb³⁺ ions. Excitation at 980 nm, under laser irradiance of 99.3 W/cm². (**d**) Compared upconversion brightness (250-850 nm) per unit volume for NaYF₄:20% Yb³⁺, 1% Tm³⁺@NaYF₄ (13.5@3.5 nm) and NaYF₄:8% Tm³⁺@NaYbF₄@NaYF₄ (3.5@5.5@3.5 nm). The ratio of UCL brightness per unit volume

for NaYF₄:8% Tm³⁺@NaYbF₄@NaYF₄ to that for NaYF₄:20% Yb³⁺, 1% Tm³⁺@NaYF₄ is marked on the top of yellow column.

Action 1. We have included Figure R1 and Figure R2 and corresponding discussions in the section of **Supplementary Note**. Moreover, we supplemented Figure R3 as **Supplementary Figure 7** in the revised manuscript. In the revised manuscript, we have added the following discussions to address the reviewer’s concern:

Page 7, paragraph 2. “We would like to note that separating Yb³⁺ and Tm³⁺ into different domains simultaneously reduces both ET efficiency and BET efficiency. While the decreased ET efficiency lowers upconversion brightness, the reduced BET efficiency mitigates concentration quenching effects, thereby enhancing brightness. This trade-off in the designed core–shell–shell nanostructure ultimately results in brighter upconversion nanoparticles, as the brightness is more significantly influenced by inhibiting the concentration quenching effect (**Supplementary Note**).”

Page 4, paragraph 1. “Note that the UCL brightness of core–shell–shell NaYF₄:8% Tm³⁺@NaYbF₄@NaYF₄ nanoparticles (~ 25 nm) is about 4.8-fold higher than that of the state-of-the-art NaYF₄:20%Yb³⁺, 1%Tm³⁺@NaYF₄ core–shell nanoparticles with an overall size of ~ 34 nm (the inert shell thickness, 3.5 nm) (**Supplementary Fig. 7**).”

The choice for numerical values of w_0 , w_1 and w_b in Supplementary Table 3 (Supplementary Table 4 in the revised SI) is as follows:

a) the energy transfer rate of Yb³⁺ (²F_{5/2}) → Tm (³H₅), w_0 , is estimated according to the corresponding value of NaYF₄:18% Yb³⁺, 0.5% Tm³⁺ (reference 2 in Supplementary Information) and relationship between energy transfer rate and doping concentration (reference 13 in Supplementary Information).

b) the energy transfer rate of Yb³⁺ (²F_{5/2}) → Tm³⁺ (³F_{2,3}), w_1 , is estimated to be 20 times as large as w_0 from de Matos, P. S. F. et al (reference 14 in Supplementary Information).

c) the back energy transfer rate of Tm³⁺ (³H₄) → Yb³⁺ (²F_{5/2}), w_b , is estimated to be one tenth of w_0 from de Matos, P. S. F. et al (reference 14 in Supplementary Information).

Action 2. We have included these discussions on selecting numerical values of w_0 , w_1 , and w_b in the revised Supplementary Table 4.

Reviewer #2 (Remarks to the Author):

The effect of lanthanide concentration quenching has long been known as a critical issue that limits the upconversion luminescence brightness of lanthanide-doped materials. In this manuscript, the authors demonstrate the inhibition of backward energy transfer from activator(Tm) to sensitizer(Yb) ions, utilizing a core–shell–shell structure, can significantly suppress this effect and increase the optimized Tm concentration from 0.5% to 8%. The conclusions look convincing and reasonably supported by properly designed experiments and quantitative results.

Therefore, I think this manuscript is suitable for publication in Nature Communications after reflecting minor revisions as indicated below.

Response: We sincerely thank the reviewer for their careful reading and positive comments on our manuscript.

1) Supplementary Figure 2a: there are two small sharp peaks located on the pattern of the 8% Tm core only (at about 38° and 56° of 2θ) which are not matching the standard ones of β -NaYF₄. Please clarify this.

Response: We thank the reviewer for the comment. These two small sharp peaks are indexed to the standard powder X-ray diffraction data of NaF (JCPDS No.36-1455), which is the by-product during the synthesis of the core nanocrystals (Figure R4).

Figure R4. Powder X-ray diffraction (XRD) patterns for 8% Tm³⁺ core, core–shell (8% Tm in the core)

and core-shell-shell samples with varying Tm^{3+} concentration in the core. Standard patterns of hexagonal phase NaYF_4 (JCPDS No.16-0334) and NaF (JCPDS No.36-1455) are included for reference. NaF is the by-product during the synthesis of the core nanoparticles.

Action. We have replaced Supplementary Figure 2a with Figure R4, and clarified the origin of the two small sharp peaks in its figure caption correspondingly.

Revised caption of Supplementary Figure 2a: Powder X-ray diffraction (XRD) patterns for 8% Tm^{3+} core, core-shell (8% Tm^{3+} in the core) and core-shell-shell samples with varying Tm^{3+} concentration in the core. Standard patterns of hexagonal phase NaYF_4 (JCPDS No.16-0334) and NaF (JCPDS No.36-1455) are included for reference. NaF is the by-product during the synthesis of the core nanoparticles.

2) Figure 2c: it seems that the optimized doping concentration is lower for higher-lying emission levels. Please explain this in detail. Also as a minor one, please replace “Integrated Intensity” with “Integrated intensity”.

Response: We thank the reviewer for the comment. We have corrected “Integrated Intensity” into “Integrated intensity” in the revised Figure 2c. The optimized doping concentration is lower for higher-lying emission levels, as the higher-lying states involved additional cross relaxation process than the lower-lying states that strongly depends on the Tm^{3+} activator concentration.

As the emission level $^1\text{G}_4$ is populated from $^3\text{H}_4$ through energy transfer upconversion from Yb^{3+} , its optimal doping concentration would theoretically match that of $^3\text{H}_4$. However, due to the presence of a cross-relaxation process ($^1\text{G}_4 + ^3\text{F}_4 \rightarrow ^3\text{F}_3 + ^3\text{H}_4$), which intensifies with increasing Tm^{3+} doping concentration. Therefore, the optimized doping concentration for $^1\text{G}_4$ is lower than that for $^3\text{H}_4$. Furthermore, the emission level $^1\text{D}_2$ is populated from $^1\text{G}_4$ by the cross-relaxation process ($^1\text{G}_4 + ^1\text{G}_4 \rightarrow ^1\text{D}_2 + ^3\text{F}_3$) and $^1\text{I}_6$ is populated from $^1\text{D}_2$ by an energy transfer upconversion process from Yb^{3+} . Consequently, the optimized doping concentration for these higher-lying emission levels ($^1\text{G}_4$, $^1\text{D}_2$, $^1\text{I}_6$) is lower than that for $^3\text{H}_4$.

Action. We have included the above discussions to Supplementary Table 2.

3) Figure 5b: please consider analysis for different emission band.

Response: We appreciate the reviewer's suggestion. The integrated intensity of the 800 nm UCL band from the $^3\text{H}_4$ state is the strongest, while the UCL intensities from the higher-lying $^1\text{G}_4$, $^1\text{D}_2$, $^1\text{I}_6$ states follow a similar trend with varying Tm^{3+} concentrations. Therefore,

in revised Figure 5b, we have separated the emission bands into two groups for discussion: one corresponding to the 800 nm UCL from the $^3\text{H}_4$ state and the other encompassing the UCL emissions from the higher-lying $^1\text{G}_4$, $^1\text{D}_2$, $^1\text{I}_6$ states (**Figure R5** below).

The overall upconversion brightness (UCL, 400-850 nm) shows a nearly linear growth with the increment of Tm^{3+} concentration up to 50%, which is about six-fold higher than the optimal Tm^{3+} concentration of 8% at laser irradiance of 99.3 W/cm^2 . The brightness of the UCL band at 800 nm exhibit a similar trend as the overall upconversion brightness with an optimal Tm^{3+} doping concentration of 50%. Note that the intensities of shorter wavelength UCL (400-760 nm) from higher-lying energy levels ($^1\text{G}_4$, $^1\text{D}_2$, $^1\text{I}_6$) (three-, four- and five-photon processes) are on par with that of UCL at 800 nm (two-photon process) under single UCNP measurement. This can be attributed to the involvement of higher-order multiphoton processes and the utilization of high laser irradiances.

Figure R5. Average single particle UCL (400-850 nm) brightness for all the core-shell-shell samples $\text{NaYF}_4:x\% \text{ Tm}^{3+}@\text{NaYbF}_4@\text{NaYF}_4$ (3.5@5@3 nm) under 980 nm laser irradiance of 20 MW/cm^2 ; note that UCL from the core-shell-shell samples with 0.5, 1 and 2% Tm^{3+} is marked by dotted box, as these samples were not bright enough to be measured. The 800 nm luminescence (760-850 nm) intensity is marked with a gray stripe pattern.

Action. We have replaced Figure 5b with Figure R5, and included the above discussions to the main text.

Page 10, paragraph 1. “The overall upconversion brightness (UCL, 400-850 nm) shows a nearly linear growth with the increment of Tm^{3+} concentration up to 50%, which is about six-fold higher than the optimal Tm^{3+} concentration of 8% at laser irradiance of 99.3 W/cm^2 . The brightness of the UCL band at 800 nm exhibits a similar trend as the overall upconversion brightness with an optimal Tm^{3+} doping concentration of 50%. Note that the intensities of shorter wavelength UCL (400-760 nm) from higher-lying energy levels ($^1\text{G}_4$, $^1\text{D}_2$, $^1\text{I}_6$) (three-, four- and five-photon processes) are on par with that of UCL at 800 nm (two-photon process) under single UCNPs measurement. This can be attributed to the involvement of higher-order multiphoton processes and the utilization of high laser irradiances.”

4) Supplementary Figure 4: please add the corresponding size distribution of samples in (a)-(g).

Response: Per the suggestion, we have added the corresponding size distribution of samples in revised Supplementary Figure 4 (see Figure R6 below).

Figure R6. TEM image of $\text{NaYF}_4:20\% \text{Yb}^{3+}, x\% \text{Tm}^{3+}$ nanoparticles at different doping levels (a) 0.2%, (b) 0.5%, (c) 1%, (d) 2%, (e) 4%, (f) 8%, (g) 16%. The corresponding size distribution is included as the inset. All nanoparticles have an average size around 27 nm. (h) Emission spectra collected under 980-nm excitation at 99.3 W/cm^2 for upconversion nanoparticles (UCNPs) in n-hexane. The absorbance of all samples has been normalized at 975 nm for the ${}^2\text{F}_{7/2} \rightarrow {}^2\text{F}_{5/2}$ transition of Yb^{3+} ions. Intensity of 800 nm emission peak multiplies by the coefficient 0.1 for better to present the data.

Action. We have replaced Supplementary Figure 4 with Figure R6.

5) Supplementary Figure 8: please mark the size of each layer for all samples.

Response: We have included the size of each layer for all samples in Supplementary Figure 8 (Supplementary Figure 6 in the revised manuscript).

Reviewer #3 (Remarks to the Author):

This study focuses on enhancing the upconversion efficiency of lanthanide-doped NaYF₄-based nanoparticles, specifically by addressing the concentration quenching effect that limits their brightness under low excitation irradiance (<100 W/cm²). The authors designed a heterogeneous core-shell-shell structure in which Tm³⁺ activator ions are confined to the core, while Yb³⁺ sensitizer ions are distributed in the inner shell. This design minimizes the back energy transfer (BET) from Tm³⁺ to Yb³⁺, a significant factor contributing to concentration quenching, and enables an increase in optimal Tm³⁺ doping concentration from 0.5% to 8%. At higher excitation irradiance (~20 MW/cm²), the optimal Tm³⁺ concentration can be pushed to as high as 50%, which is unprecedented for single-particle systems. This work provides new insights into enhancing upconversion nanoparticle (UCNP) efficiency and lays the groundwork for broader applications in fields like biosensing, bioimaging, and solid-state lasing.

Therefore, I recommend it for publication after addressing the minor points:

Response: We sincerely appreciate the reviewer's positive feedback and valuable suggestions on our work.

1. While the study focuses on Yb-Tm systems, it would be beneficial to discuss whether the same architecture could be applied to other lanthanide pairs, such as Yb-Er or Yb-Ho, which are also commonly used in upconversion applications.

Response: We thank the reviewer's advice. We have performed additional experiments, and demonstrated that the same core-shell-shell architecture can be extended to Yb³⁺-Er³⁺ and Yb³⁺-Ho³⁺ upconversion systems, accommodating higher activator concentrations for bright upconversion (Figure R7).

Action. We have added Figure R7 as Supplementary Figure 8 in the revised SI, and added corresponding discussions to the revised main text.

Page 4, paragraph 1. “Moreover, the designed core-shell-shell architecture can also be extended to the commonly used Yb³⁺-Er³⁺ and Yb³⁺-Ho³⁺ upconversion systems to accommodate high activator concentrations for bright upconversion (Supplementary Fig. 8)”

Figure R7. Upconversion luminescence spectra collected under 980 nm laser irradiance at 99.3 W/cm² for (a) NaYF₄:2% Er³⁺@NaYbF₄@NaYF₄ (3.5@5@3 nm) and NaYF₄:8% Er³⁺@NaYbF₄@NaYF₄ (3.5@5@3 nm), (c) NaYF₄:2% Ho³⁺@NaYbF₄@NaYF₄ (3.5@5@3 nm) and NaYF₄:8% Ho³⁺@NaYbF₄@NaYF₄ (3.5@5@3 nm) nanoparticles in n-hexane. The absorbance of all samples has been normalized at 975 nm for the ²F_{7/2} → ²F_{5/2} transition of Yb³⁺ ions. Simplified energy levels of Yb³⁺/Er³⁺ (b) and Yb³⁺/Ho³⁺ (d) shows energy transfer upconversion and back energy transfer (BET) processes under 980 nm laser excitation. Solid line with an arrow, absorption or emission; curve with an arrow, multiphonon relaxation; dashed line with an arrow linked by dotted line, non-radiative energy transfer.

2. The long-term stability of the core-shell-shell structure under extended irradiation conditions was not discussed in detail. Information on the stability of the designed architecture would further enhance the impact of the study.

Response: We thank the reviewer's advice for discussion on long-term stability of the designed core-shell-shell structure under extended irradiation conditions. We recorded upconversion luminescence intensity trajectory of a single nanoparticle for the brightest sample β-NaYF₄:50% Tm@NaYbF₄@NaYF₄ (3.5@5@3 nm) at 20 MW/cm² irradiance of

980 nm laser. The result indicates no noticeable photo-bleaching for the core-shell-shell UCNPs during measurements over 1 hour (Figure R8). Such an excellent photostability is highly desired for biological studies based on confocal microscopy.

Figure R8. The time trace of upconversion luminescence (UCL, 400-850 nm) intensity from a single nanoparticle for NaYF₄:50% Tm³⁺@NaYbF₄@NaYF₄ (3.5@5@3 nm) under continuous 980-nm excitation at 20 MW/cm². The bin time for each data point is 10 ms.

Action. We have included Figure R8 as Supplementary Figure 14 in the revised SI, and added the corresponding discussion in the revised main text.

Page 10, paragraph 1. “No noticeable UCL photo-bleaching or photo-blinking was observed over 60 mins single nanoparticle measurements, affirming the stable structure of the designated core-shell-shell UCNPs at high laser irradiances (**Supplementary Fig. 14**)”